# Phylogenetic and Morphological Characteristics Reveal Cryptic Speciation in Stingless Bee, *Tetragonula laeviceps s.l.* Smith 1857 (Hymenoptera; Meliponinae)

**DOI:** 10.3390/insects14050438

**Published:** 2023-05-03

**Authors:** Kimberly Ador, Januarius Gobilik, Suzan Benedick

**Affiliations:** Faculty of Sustainable Agriculture, Universiti Malaysia Sabah, Locked Bag No. 3, Sandakan 90509, Malaysia

**Keywords:** body size, body color, COI mtDNA, intraspecific variation, *Tetragonula laeviceps*

## Abstract

**Simple Summary:**

The status of DNA barcoding in the cryptic species of stingless bees from Borneo, *Tetragonula laeviceps sensu lato (s.l.)* (Smith 1857), is poorly known. The *T. laeviceps s.l.* samples used in this study, which contain worker bee individuals grouped according to morphological characteristics and morphometric variations, could potentially have a similar grouping of COI haplotypes, but this has not yet been investigated. In this study, we investigate whether individuals of *T. laeviceps s.l.* worker bees grouped according to the same or nearly the same morphological traits show similar COI haplotype cluster groupings. The specimens were first classified according to the most obvious morphological characteristics, i.e., hind tibia color, hind basitarsus color and body size and grouping was based on morphological characteristics important for distinguishing the four groups within *T. laeviceps s.l*. The most distinctive features of the morphological and morphometric characteristics measured by PCA and LDA biplots that distinguished Group 1 from the other groups were the blackish-brown antennae scape (ASC) and the black (TC). Group 2 had a yellowish-brown ASC and a dark TC, while Group 3 had a dark brown ASC; a black TC; and the largest TL, FWW, and FWL. As for phylogenetic relationships, 12 out of 36 haplotypes showed clear separation with good bootstrap values (97–100%). The remaining haplotypes did not show clear differentiation between subclades that belonged together, regardless of their morphology and morphometric characteristics. Thus, the combination of DNA barcoding for species identification and phylogenetic analysis, as well as traditional methods based on morphological grouping by body size and body color, can be reliably used to determine intraspecific variations, such as the possible occurrence of subspecies within *T. laeviceps s.l*.

**Abstract:**

*Tetragonula laeviceps sensu lato (s.l.)* Smith 1857 has the most complicated nomenclatural history among the *Tetragonula* genera. The objective of this study was to investigate whether *T. laeviceps s.l.* individuals with worker bees are grouped in the same or nearly the same morphological characteristics and have similar COI haplotype cluster groups. A total of 147 worker bees of *T. laeviceps s.l.* were collected from six sampling sites in Sabah (RDC, Tuaran, Kota Marudu, Putatan, Kinarut and Faculty of Sustainable Agriculture (FSA)), but only 36 were selected for further studies. These specimens were first classified according to the most obvious morphological characteristics, i.e., hind tibia color, hind basitarsus color and body size. Group identification was based on morphological characteristics important for distinguishing the four groups within *T. laeviceps s.l*. The four groups of *T. laeviceps s.l.* had significantly different body trait measurements for the TL (total length), HW (head width), HL (head length), CEL (compound eye length), CEW (compound eye width), FWLT (forewing length, including tegula), FWW (forewing width), FWL (forewing length), ML (mesoscutum length), MW (mesoscutum width), SW (mesoscutellum width), SL (mesoscutellum length), HTL = (hind tibia length), HTW (hind tibia width), HBL (hind basitarsus length) and HBW (hind basitarsus width) (*p* < 0.001). Body color included HC (head color), CC (clypeus color), ASC (antennae scape color), CFPP (Clypeus and frons plumose pubescence), HTC (hind tibia color), BSC (basitarsus color), SP (leg setae pubescence), SP (Thorax mesoscutellum pubescence), SPL (thorax mesoscutellum pubescence length) and TC (thorax color) (*p* < 0.05). The most distinctive features of the morphological and morphometric characteristics measured by PCA and LDA biplot that distinguish Group 1 (TL6-1, TL6-2 and TL6-3) from the other groups were the yellowish-brown ASC and the dark brown TC. Group 2 (haplotypes TL2-1, TL2-2 and TL2-3 and TL4-1, TL4-2 and TL4-3) had a dark brown ASC and a black TC, while Group 3 (haplotypes TL11-1, TL11-2 and TL11-3) had a blackish-brown ASC, a black TC and the largest TL, FWW and FWL. As for phylogenetic relationships, 12 out of 36 haplotypes showed clear separation with good bootstrap values (97–100%). The rest of the haplotypes did not show clear differentiation between subclades that belonged together, regardless of their morphology and morphometric characteristics. This suggests that the combination of DNA barcoding for species identification and phylogenetic analysis, as well as traditional methods based on morphological grouping by body size and body color, can be reliably used to determine intraspecific variations within *T. laeviceps s.l*.

## 1. Introduction

*Tetragonula laeviceps sensu lato (s.l.)* (Smith, 1857) is widely distributed in the Indo-Malayan region [1,2,3,4,5,6,7,8,9,10,11,12,13,14,15]. This species differs from other species of the *Tetragonula* genus by having a predominantly shiny black body, longer wing length, a black mesoscutum, a metasoma that is completely covered with yellowish setae [1] and a hind wing with five hamuli [8,9,10]. According to Rasmussen and Michener [11], *Tetragonula laeviceps s.l.*, formerly known as *Trigona laeviceps s.l.* (Smith, 1857), has the most complicated nomenclatural history among the genera of Tetragonula. Previously, Schwarz [12] considered specimens of this species to be *Tetragonula iridipennis* (Smith 1854), but later studies showed that *T. iridipennis* specimens are restricted to India and Sri Lanka, while other specimens of *T. iridipennis* identified by Schwarz in other areas were renamed *T. laeviceps s.l.* by Sakagami [7]. This species is distinguished from *T. iridipennis* by its mesoscutal hairs, which are not well banded, and the greater length of the wings [7]. A recent study by Siti Fatimah et al. [13] found that *T. laeviceps s.l.* can be distinguished from other genera by a blackish-brown mandible, a black basal color and a long band in the middle of the mesoscutum that is not flanked by other bands.

Many organisms in tropical areas, such as the stingless bee, are still awaiting discovery due to their cryptic morphology, i.e., the lack of reliable morphological differences between genera and within species. Attempts have been made to classify stingless bees based on body size, number of hamuli, forewing and hind wing length and cephalic characteristics, among other characteristics [16,17,18]. To date, researchers have not reached a consensus on the taxonomic status of the cryptic species *T. laeviceps s.l.*, which exhibits strong intraspecific variations with regard to morphological characteristics [1,4,11,12,13,14,15,16]. For example, the body size of *T. laeviceps s.l.* workers varies from region to region. It was measured in previous studies in Indonesia (Sumatra), Singapore and Peninsular Malaysia (Selangor) by Abu Hassan [4], who found measurements ranging from 3.5 to 3.99 mm. In Peninsular Malaysia, Siti Fatimah et al. [13] measured values from 4.75 to 5.41 mm; in Indonesia (Java), Purwanto et al. [14] measured values from 3.42 to 3.45 mm; in Indonesia (Yogyakarta), Trianto and Purwanto [8] measured values from 3.44 to 3.76 mm; and in Indonesia (Sulawesi), Suriawanto [19] measured body sizes ranging from 3.40 to 3.43 mm. Furthermore, there is significant color variation in the body parts of the *T. laeviceps s.l.* species found in Malaysia, such as the mesoscutellum, and hairs on the mesoscutellum, clypeus, tegulae, head, antennae and legs, as studied by Abu Hassan [4] and Siti Fatimah et al. [13]. Therefore, this study expects that there will be a distinct grouping of morphological characteristics in individuals of the *T. laeviceps s.l.* species. To achieve this, worker bees from different districts in the state of Sabah, Malaysia, were collected and matched to individuals with the same or similar morphological and morphometric characteristics of the same group and examined for a taxonomic analysis.

Over the last few decades, DNA barcoding [20] has emerged as an important genetic tool for the taxonomic identification of bee species [21,22,23,24,25] that were difficult to morphologically identify and/or did not have dichotomous keys [26,27]. To date, most sequences have been obtained from the Meliponini cytochrome oxidase I (COI) barcode fragment to support morphometric data and address specific taxonomic questions [18,28,29,30,31,32,33,34,35,36,37] or conduct phylogenetic and phylogeographic research [38,39,40]. DNA barcoding with the COI gene of mitochondrial DNA (mtDNA) is a protein-coding gene that contains more differences than the ribosomal gene [21]; therefore, it can be used to distinguish closely related species in addition to the morphological characteristics and morphometric variations of insects. To date, the status of DNA barcoding for the cryptic species of stingless bees from Borneo, *T. laeviceps s.l.*, is poorly known. There is a possibility that *T. laeviceps s.l.* samples in this study, which contain worker bee individuals grouped in a particular set of morphological characteristics and morphometric variations, have a similar grouping of COI haplotypes, but this has not yet been investigated.

## 2. Materials and Methods

### 2.1. Stingless Bee Samples

A total of 147 *T. laeviceps s.l.* workers were collected directly from the nest entrance of the colony at six sampling sites in Sabah, including RDC, Tuaran, Kota Marudu, Putatan, Kinarut and the Faculty of Sustainable Agriculture (FSA) at Universiti Malaysia Sabah. The stingless bees were kept in collection bottles, stored in 100% ethanol and taken to FSA Sandakan for molecular analysis. The voucher specimens were deposited in the insect collection in the FSA herbarium. However, only 36 *T. laeviceps s.l.* individuals were selected for further systematic studies. These specimens were first classified according to the most obvious morphological characteristics, namely body size, hind tibia color and hind basitarsus color, which can be detected by eye as an important morphological feature, to create four distinct groups within *T. laeviceps s.l.* (Table 1). Additional *T. laeviceps s.l.* individuals with similar characteristics were excluded from this study.

To ensure that the species obtained in this study most closely resembled the morphological characters of *T. laeviceps s.l.,* we also collected four species of the genus Tetragonula as comparative species, namely *Tetragonula melanocephala s.l.* (i. GenBank Accession Nos. ON459535 (total length (TL) = 5.20 mm and mesoscutum width (SW): 1.30 mm) and ii. ON459537 (TL = 5.23 mm and SW = 1.33 mm)), *Tetragonula fuscobalteata s.l.* (GenBank Accession No. i. ON458748 (TL = 3.16 mm and SW = 0.94 mm) and ii. ON458746 (TL = 3.13 mm, and SW = 0.97 mm)), *Tetragonula ruficornis s.l.* (GenBank Accession No. ON458756, TL = 3.45 mm, SW = 1.07 m) and *Tetragonula iridipennis s.l.* (GenBank Accession No. i. ON458758 (TL = 3.54 mm, SW = 1.01) and ii. ON458757 (TL = 3.57 mm, SW = 1.05 mm)), but they were excluded from the present analysis. *Tetragonula fuscobalteata s.l.*, which was used as an outgroup species for the phylogenetic analysis, was described by the smallest size, six distinct longitudinal pubescence and five glabrous areas on the mesoscutum and reduced submarginal of the forewing cells (see Figure A1) [4].

### 2.2. Morphological Characteristics and Morphometric Parameters of T. laeviceps s.l.

In this study, the overall measurements of the morphological characteristics of *T. laeviceps s.l.* were carried out as follows:

#### 2.2.1. Body Size

In this study, 16 measurements of the body size of *T. laeviceps s.l.* were selected as quantitative morphometric parameters (Table 2), following Suprianto et al. [37], Trianto and Purwanto [8], Siti Fatimah et al. [13], Smith [9], Dollin et al. [38], Sakagami et al. [39], Sakagami and Inoue [40] and Sakagami [7]. All *T. laeviceps s.l.* workers were photographed under a Leica DFC495 microscope (Danaher Corporation, Washington, DC, USA) at 10 to 20× magnification. In each individual, the following morphological characteristics were measured: total length (TL), head width (HW), head length (HL), compound eye length (CEL), compound eye width (CEW), forewing length (including tegula) (FWLT), forewing width (FWW), forewing length (FWL), mesoscutum length (ML), mesoscutum width (MW), mesoscutellum width (SW), mesoscutellum length (SL), hind tibia length (HTL), hind tibia width (HTW), hind basitarsus length (HBL) and hind basitarsus width (HBW).

#### 2.2.2. Body Color and Pubescence

Eleven (11) qualitative measurements of body pubescence and color of head, antennae and thorax were used in this study: head color (HC), clypeus color (CC), antennae scape color (ASC), clypeus and frons plumose pubescence (CFPP), hind tibia color (HTC), hind basitarsus color (BSC), hind leg setae pubescence (LSP), hind leg setae pubescence length (LSPL), thorax mesoscutellum pubescence (SP), thorax mesoscutellum pubescence length (SPL) and thorax color (TC) (Table 2). The measurements of the body color and pubescence of *T. laeviceps s.l.* were based on the quantitative color code described by Abu Hassan [4] and Sakagami et al. [39]: (i) body color (1—black, 2—blackish-brown, 3—dark brown and 4—yellowish-brown) and (ii) body pubescence (1—sparse and 2—dense).

### 2.3. Molecular Analyses of T. laeviceps s.l.

#### 2.3.1. DNA Extraction and PCR Amplification

To detect a possible differentiation of the mtDNA haplotypes, a total of 37 stingless bee individuals, comprising 36 *T. laeviceps s.l.* workers and 1 worker of *Tetragonula fuscobalteata* (outgroup species), were examined for molecular analyses (Table 1). As indicated by Koch [33], DNA was extracted from the entire body of the stingless bee using the Wizard Genomic DNA Extraction Kit (Promega, Madison, WI, USA), as the thorax alone did not yield enough DNA for a subsequent analysis. The entire body of the stingless bee was determined to be the best source of mitochondrial DNA [41]. The targeted cytochrome c oxidase I (COI) primers LepF1 (forward) 5′-ATTCAACCAATCATAAAGATATTGG-3′ and LepR1 (reverse) 5′ TAAACTTCTG GATGTCCAAAAAATCA-3′ were used for PCR amplification and DNA sequencing [33].

#### 2.3.2. Sequence Editing and Alignment

All analyses were performed on the 680-nucleotide portion of the alignment of 37 sequences. The quality of the reads was checked using Chromas 2.6.6 (Technelysium DNA Sequencing Software). Raw sequence reads were processed using Mega X version 10.0.5 [42] and aligned using Clustal X [43]. Additionally, Mega X version 10.0.5 was used to assemble and edit the contigs (the consensus sequence from the region of overlap between the 5’ and 3’ sequences). The sequences were then imported into DnaSP v6 [44] to generate haplotypes. Clustal X and MEGA X version 10.0.5 were utilized to align the sequences with the *T. fuscobalteata* outgroup.

### 2.4. Data Analysis

#### 2.4.1. Analysis of the Morphological and Morphometric Parameters of *T. laeviceps s.l.*

Univariate (one-way ANOVA and Kruskal–Wallis ANOVA), multivariate (PCA and LDA) and matrix correlation analyses of the morphological and morphometric parameters were performed using SPSS version 26.0 and Past 4.03. Qualitative data were converted to log10 for further parametric analyses. To identify meaningful morphological and morphometric characteristics that can differentiate the four *T. laeviceps s.l.* groups, 27 variables were analyzed using the PCA analysis. The values (scores) for each principal component were obtained from PCA scree plot eigenvalues > 1. The contributions of the variables (correlation values) to each principal component (PCA) were interpreted as informative if >0.60.

#### 2.4.2. Correlations, PCA and LDA Biplot Analysis

The PCA biplot analysis used variables of the correlation matrix of 21 morphological and morphometric traits and the calculation matrix of the four *T. laeviceps s.l.* groups. All correlations between the main axes, the vectors of the variables (loadings) and the *T. laeviceps s.l.* groups (scores) are presented in the PCA biplot. To obtain a correct predictive classification between the groups of *T. laeviceps s.l.*, the sample in the LDA was split from a dataset of the variables identifying a validation set of 3 groups of traits based on the selection algorithm: (i) A (TL, HW, HL, CEL, CEW, FWLT, FWW, FWL, ML, MW, SW, SL, HTL, HTW, HBL and HBW); (ii) B (CC, ASC and TC) and (iii) C (ML and CFPP). In this study, LDA used a PCA-based multivariate evaluation where the resulting canonical axes (Axis 1 and Axis 2) showed the largest separation between all groups.

#### 2.4.3. Sequence and Phylogenetic Analysis

The maximum likelihood method (ML) was used to reconstruct the phylogenetic relationships between the *T. laeviceps s.l.* haplotypes. For phylogenetic reconstruction, a ML optimality criterion (as implemented in Paup 4.0a169 [45]) was used with the Kimura-2 parameter (K2P) model and *T. fuscobalteata* as the outgroup. An optimal model of nucleotide evolution of the ML analysis was determined using ModelTest 2 [46]. Branch swapping was performed through Tree Bisection Reconnection (TBR). An estimate of support for each branch was determined with the bootstrap test using 1000 pseudoreplications.

## 3. Results

### 3.1. Identification of T. laeviceps s.l.

In Figure 1, the identification of *T. laeviceps s.l.* workers were confirmed; the main features include four longitudinal hairs that are not well banded, three glabrous areas of the mesoscutum, reduced submarginal cells of the forewing and the elliptical disc of the basitarsus, as described by Abu Hassan [4], Purwanto et al. [14], Suriawanto et al. [47], Siti Fatimah et al. [13] and Sakagami [7]. In the morphometric analyses, four groups with different morphological characteristics were found among the 36 *T. laeviceps s.l.* specimens, and further morphometric and DNA barcoding analyses were performed on them. Additionally, all 36 specimens were matched with BOLD private entries of *T. laeviceps s.l.* (% similarity = 75.4–100) (Table 1).

### 3.2. Morphological and Morphometric Parameters Analysis of T. laeviceps s.l.

#### 3.2.1. Quantitative Morphological Characteristics

Overall, the total body length of *T. laeviceps s.l.* was highest in Group 3 (mean TL = 4.24 mm), followed by Group 4 (mean TL = 3.74), Group 2 (mean TL = 3.43 mm) and Group 1 (mean TL = 3.43 mm) (Table 3). One-way ANOVA between the four *T. laeviceps s.l.* groups showed that the body traits measurements were significantly different for the TL, HW, HL, CEL, CEW, FWLT, FWW, FWL, ML, MW, SW, SL, HTL, HTW, HBL, and HBW (*p* < 0.001) (Table 3).

#### 3.2.2. Qualitative Analysis of Body Color and Pubescence

The Kruskal–Wallis test for body color and pubescence between the four groups revealed that the HC, CC, ASC, CFPP, HTC, BSC and TC were significantly different at *p* < 0.001, while LSP, SP and SPL were significantly different at *p* < 0.05. However, a nonsignificant difference was detected for the LSPL (Table 4; *p* > 0.05).

#### 3.2.3. PCA and LDA Biplot Analysis

The PCA biplot in Figure 2 displays the correlation between the principal axes (PC1 and PC2), 20 morphological and morphometric parameters (loadings) and *T. laeviceps s.l.* groups (PCA scores). The first two principal components (PC1 and PC2) describe 91.56% of the total variance and have eigenvalues greater than the unit. Small angles indicate a strong correlation between the morphological and morphometric parameters. Thus, the group of parameters based on PCA are as follows: (i) A (body sizes; TL = total length, HW = head width, HL = head length, CEL = compound eye length, FWLT = forewing length (+tegula), FWW = forewing width, FWL = forewing length, MW = mesoscutum width, SW = mesoscutellum width, SL = mesoscutellum length, HTL = hind tibia length, HTW = hind tibia width, HBW = hind basitarsus width, and HBL = hind basitarsus length), (ii) B (compound eye width (CEW) and body color (CC = clypeus color, ASC = antennae scape color and TC = thorax color) and (iii) C (mesoscutum length (ML) and clypeus and frons plumose pubescence (CFPP)).

The PCA biplot in Figure 2 indicates the correlation between the principal axes (PC1 and PC2), 21 morphological and morphometric parameters (loadings) and the *T. laeviceps s.l.* groups (PCA scores). The first two principal components (PC1 and PC2) describe 91.14% of the total variance and have eigenvalues greater than the unit. Small angles indicate a strong correlation between the morphological and morphometric parameters. In the PCA biplot, the four groups of *T. laeviceps s.l.* specimens are represented by convex hulls (Groups 1, 2, 3 and 4) that are completely separated from each other (Figure 2). Compared to the other groups, Groups 3 and 4 are the most closely related groups with the least differences in morphological and morphometric parameters in terms of body size, body color and pubescence; in these two groups, most endpoints of the respective parameters coincide. Groups 3 and 4 also have a larger body size than groups 1 and 2, and both groups have dark brown ASC and black TC. However, Groups 1 and 2 have the most distinctive features of the overall group parameters, with Group 1 having the most distinctive features of large CEW, as well as blackish brown ASC and black TC, and Group 2 having the most distinctive features of relatively larger sizes of FWLT and FWL although it is the third smallest after groups 3 and 4, as well as yellowish-brown ASC and dark brown TC. 

The linear discriminant analysis (LDA) determines the group means and calculates the probability of each specimen belonging to different groups (Figure 3). Based on the PCA multivariate analysis, the LDA also showed a clear separation of the four groups between canonical axes 1 and 2, and Groups 3 and 4 are also more closely related.

#### 3.2.4. Sequence and Phylogenetic Analysis

The mtDNA of 36 similar specimens of *T. laeviceps s.l.* from Sabah (Tuaran, Kinarut, Putatan, Kota Marudu, FSA, RDC and Beluran), analyzed for their morphological and morphometric characteristics, was amplified with an average size of 623 bp. It was found that 456 (73.19%) of the 623 traits were constant in the sequences, and 167 (26.81%) variable traits remained. Of the 623 variable features, 94 sites were parsimony informative characteristics (56.29%); the remaining 43.71% were parsimony uninformative. The probability values for the intraspecific divergence of *T. laeviceps s.l.* ranged from 26% to 100% (Figure 4). However, the *T. laeviceps s.l.* specimens showed clear separation with good bootstrap values (97–100%) for TL11-1, TL11-2 and TL11-3 from Group 3; TL6-1, TL6-2 and TL6-3 from Group 1 and TL2-1, TL2-2 and TL2-3 and TL4-1, TL4-2 and TL4-3 from Group 2 (Figure 4). The rest of the specimens displayed no clear distinction between the subgroups, which belonged together regardless of their morphology and morphometric characteristics.

The morphological and morphometric characteristics described in Table 5 support a possible separation of haplotypes between Group 1, Group 2 and Group 3 in Figure 4. The most pronounced features in Group 1 (haplotypes TL6-1, TL6-2 and TL6-3) that distinguished it from the other groups were the blackish-brown ASC and the black TC. Group 2 (haplotypes TL2-1, TL2-2 and TL2-3 and TL4-1, TL4-2 and TL4-3) had a yellowish-brown ASC and a dark-brown TC, while Group 3 (haplotypes TL11-1, TL11-2 and TL11-3) had a dark-brown ASC; a black TC and the largest TL, FWW and FWL (Table 5).

## 4. Discussion

### 4.1. Distinctive Morphological and Morphometric Parameters of T. laeviceps s.l. Based on Body Size, Color and Hair

In this study, the indeterminacy of morphological differences within workers of *T. laeviceps s.l.* was observed, with differences in body size (Table 3; 16 traits, *p* < 0.001) and in body color and pubescence (Table 4; 10 out of 11 traits, *p* < 0.05). The morphological traits of the four groups of *T. laeviceps s.l.* workers differed in size (see four groups of *T. laeviceps s.l.* in Table 1). A similar finding was also reported by Purwanto and Trianto [15] in Indonesia and Patel and Pastagia [48] in India. Table 3 shows that the worker bees in Group 3 had the greatest mean value of the total length (TL = 4.24 mm), head width (HW = 1.82 mm), head length (HL = 1.55 mm), compound eye length (CEL = 1.28 mm), compound eye width (CEW = 0.46 mm), forewing length (including tegula) (FWLT = 4.39 mm), forewing width (FWW = 1.52 mm), forewing length (FWL = 3.99 mm), mesoscutum width (MW = 1.18 mm), mesoscutum length (MW = 1.05 mm), mesoscutellum width (SW = 0.78 mm), mesoscutellum length (SL = 0.31 mm), hind tibia length (HTL = 1.69 mm), hind tibia width (HTW = 0.59 mm), hind basitarsus length (HBL = 0.66 mm) and hind basitarsus width (HBW = 0.33 mm). The second-largest workers were from Group 4, which had a mean total length of 3.74 mm, head width of 1.66 mm, head length of 1.47 mm, compound eye length of 1.25 mm, compound eye width of 0.43 mm, length of forewing (including tegula) of 4.34 mm, forewing width of 1.42 mm, forewing length of 3.94 mm, mesoscutum width of 1.14 mm, mesoscutellum width of 0.74 mm, mesoscutellum length of 0.24 mm, length of hind tibia of 1.57 mm and length of hind basitarsus of 0.62 mm. The smallest workers were from group 1, which had a TL of 3.43 mm, HW of 1.54 mm, HL of 1.38 mm, CEL of 1.03 mm, CEW of 0.45 mm, FWLT of 3.69 mm, FWW of 1.21 mm, FWL of 3.30 mm, ML of 0.95 mm, MW of 1.06 mm, SW of 0.66 mm, SL of 0.21 mm, HTL of 1.36 mm, HTW of 0.59 mm, HBL of 0.55 mm and HBW of 0.25 mm. Overall, the *T. laeviceps s.l.* bees with the greatest mean total length also had greater mean values in the other morphological characteristics.

Previous studies by May-Itzá et al. [18], Novita et al. [49], Quezada-Euán et al. [50] and Raffiudin et al. [51] found that the availability of food and nesting resources in the environment affects the size of the morphological features in stingless bees, such as body segments. As shown in Table 3, the largest mean morphological traits of *T. laeviceps s.l.* worker bees in relation to size for Group 3 are from the agricultural area where many oil palms were planted, followed by Group 4 from the natural and secondary forest, Group 2 from the urban vegetation and Group 1 from the secondary forest and teak plantation (Table 1). Changes in temperature or environmental conditions cause organisms to morphologically adapt to the environment, including flight and foraging activities [51]. This was further confirmed by Roubik [52] and Ruttner [53], who found that body size variations in worker bees are a form of adaptation in the foraging and exploitation of floral resources. Foraging distances also depend on environmental conditions, such as the density and distribution of food resources and the general physical carrying capacity of different habitats [54,55]. Larger bees with bigger thoraces tend to have greater flight distances, allowing them to access more extensive areas for food and nesting material sources from the environment [51,52,53,54,55,56]. Therefore, variations in some traits, such as the length of wings and legs, may be an important factor in the morphometry of stingless bees and can be used to estimate size differences between worker bees [57,58]. Furthermore, Sun and Chaplin-Kramer [55] found that the habitat suitability for pollinators is altered by changes in forest and pasture covers, which affect the morphological characteristics of stingless bees.

For the traits of body color and pubescence in *T. laeviceps s.l.* workers, 10 out of 11 morphometric traits differed significantly between the groups (Table 4; *p* < 0.05). In this study, the four groups can be clearly distinguished from each other on the basis of the following characteristics: head color (HC), clypeus color (CC), antennal color (ASC), clypeus and frons plumose pubescence (CFPP), hind tibia color (HTC), basitarsus color (BSC), leg setae pubescence (LSP), mesoscutellum pubescence (SP), mesoscutellum pubescence length (SPL) and thorax color (TC) (see four groups of *T. laeviceps s.l.* in Table 1 and Table 4). Group 1 had a black HC, dark brown CC, blackish-brown ASC, black HT and BSC, dense LSP and SP, long LSPL and SPL and black TC. Group 2 can be distinguished by a blackish-brown HC, yellowish-brown CC and ASC, dark brown HT and BSC, sparse LSP and SP, short LSPL and SPL and dark brown TC. Group 3 showed a black HC, dark brown CC and ASC, blackish-brown HT and BSC, sparse LSP and SP, short LSPL and SPL and black TC. Group 4 had a black HC, dark brown CC and ASC, blackish-brown HT and BSC, sparse LSP and SP, short LSPL and SPL and black TC. Table 4 shows that the most obvious criterion for distinguishing the four groups by eye is the color of the hind legs (HTC) and the basitarsus (BSC). The worker bees have corbiculates, a specialized structure on their hind legs that is used to collect pollen and resin [59]. Therefore, the variations in the morphological characteristics of the hind legs of stingless bees, such as the color and shape of the corbiculae in this study, could be due to their coevolution with angiosperms and gymnosperms as pollinators [60,61]. Bomtorin et al. [61] found that the development of hind leg diphenism, characteristic of corbiculae bee species, is driven by several caste-preferentially expressed genes, such as those encoding cuticular protein genes, P450 and Hox proteins, in response to the naturally diverse diet offered to honeybees during the larval period. As for the other characteristics of body color and hair of the *T. laeviceps s.l.* worker group, it was difficult to distinguish between them. Even though two of the three groups generally had the same color and hair patterns, these groups differed statistically, and therefore, further research is needed.

### 4.2. PCA and LDA Biplot and Phylogenetics Relationships of T. laeviceps s.l.

The principal component analysis (PCA) is a commonly used analytical technique in taxonomic research, because it can identify the role of each trait in each group formed [62,63]. In this study, both PCA and LDA biplots showed a complete separation of morphological and morphometric traits (variable Group A: HW, CEL, FLT, FW, FWL, MW, SW, HTL, HTW, HBL, HBW and ASC; variable Group B: TL, HL, CEW, SL, SPL and TC and variable Group C: CFPH) between groups 1, 2, 3 and 4 of *T. laeviceps s.l.* (Figure 2 and Figure 3). However, Groups 3 and 4 had the most distinct group parameters, followed by Groups 1 and 2, indicating that the cryptic Bornean species *T. laeviceps s.l.* has morphological sizes and morphometric characteristics that are difficult to distinguish by eye and vary between habitats. This species occurs in a wide range of habitats in the Malaysian state of Sabah, from urban areas to forests and agricultural fields, and it can generally disperse over short distances. The results of this study suggest that the geological structure and diversity of tropical vegetation in Sabah, which may have been an important barrier to dispersal between populations (Table 1), is the most likely explanation for the observed patterns of variation in the morphological and morphometric characteristics of *T. laeviceps s.l*.

Of the 36 haplotypes, only 12 haplotypes from Group 3 (TL11-1, TL11-2 and TL11-3); Group 2 (TL2-1, TL2-2 and TL2-3 and TL4-1, TL4-2 and TL4-3) and Group 1 (TL6-1, TL6-2 and TL6-3) of *T. laeviceps s.l.* showed clear separation with good bootstrap values (97–100%) (Figure 4). The other haplotypes did not show clear differentiation between the subclades, which belonged together regardless of their morphology and morphometric characteristics. In this study, the most important characteristics to distinguish between the groups of *T. laeviceps s.l.* were the color of the antennal scape (ASC) and thorax (TC), forewing length (FWL) and forewing width (FWW) (Table 5). The result is further supported by the PCA and LDA biplots, which show that Groups 3 and 4 also have larger body size than groups 1 and 2, and both groups have dark brown ASC and black TC. Group 1 has a relatively large CEW, although it has the smallest body size, as well as blackish brown ASC and black TC, and Group 2 has a relatively larger FWLT and FWL, although it is the third smallest after groups 3 and 4, as well as yellowish-brown ASC and dark brown TC. Invertebrate species with low dispersal ability, such as bees and wasps, tend to be more affected by physical barriers that act as obstacles to movement between habitats, thereby affecting the community composition, leading to low or limited genetic exchanges [64,65]. Genetic differentiation can occur during evolution even if the morphological traits do not change [66,67]. However, certain external morphological traits may be retained due to selection pressure on important adaptive traits [68]. This study found that cryptic and polymorphic characteristics within species can complicate traditional taxonomic research. Both issues are common in communities of stingless bees such as *T. laeviceps s.l.* and deserve more attention during systematic identification.

The color of the hind tibia (HT), hind basitarsus (BSC), antennal scape (ASC) and the thorax (TC) are the most significant morphological features that distinguish the three groups of *T. laeviceps s.l.* (Table 1 and Table 5, Figure 4). According to Castanheira and Contel [69], the color of the mesepisternum distinguishes the two subspecies of *Tetragonisca angustula angustula* and *Tetragonisca angustula fiebrigi*. In some cases, interbreeding between the species could also result in hybrids with different mesepisterna colors [70,71,72]. This could be related to differences in the spatial regulation of the expression of other genes during development, such as the Hox genes, which, in turn, lead to morphological differences within and between taxa [73]. Jeong et al. [74] found that the decline in male abdominal pigmentation in the lineage of *Drosophila santomea* was related to the selective loss in expression of both light brown and yellow pigmentation genes in the posterior abdominal segments of males. Previous studies have also demonstrated that changes in the expression and function of Hox genes are key to the evolution of novel body structures in insects, such as segmental differences within the body and thoracic legs in some Orthoptera and Hemiptera species, and possibly abdominal appendages in Ephemeroptera [75,76]. Averof [76] found that Hox gene mutations can convert characteristic structures of one segment into the corresponding structures of another segment. Furthermore, the gene responsible for bee coloration in this study could be determined by the F1 generation, which is expressed in both queens and worker bees and confers a similar phenotype [77,78,79].

Another important feature of *T. laeviceps s.l.* identified by PCA and biplot analyses is the size of the FWL and FWW (Table 3, Figure 2, Figure 3 and Figure 4). In this study, the bees with the largest mean total body length also had higher mean values for other morphological characteristics by body size. Group 3 is the largest of the *T. laeviceps s.l.* groups, followed by Group 2 and Group 1 (Table 5). Group 2 is characterized by relatively large FWLT and FWL values, although it is the third smallest after Groups 3 and 4 (Table 3, Figure 2). Rasmussen and Michener [11] also pointed out that the earliest studies of *T. laeviceps s.l.* by Sakagami [7] also showed that different body sizes can be considered as different species, but further systematics studies are needed. In Guatemala and Mexico, morphometric analyses based on body size in combination with molecular tools (DNA barcoding) have been used to identify complex species of the stingless bee *Melipona yucatanica* [34,36]. However, Figure 4 does not show a clear distinction between the subgroups of some haplotypes of *T. laeviceps s.l.*, which belong together regardless of their morphology and morphometric characters. Based on this study, the morphologically cryptic species *T. laeviceps s.l.* needs to be further characterized, as the insects found on the islands may have different speciation processes [80].

In addition, morphometric measurements of the hymenopteran forewing (length and width of the wing, length of marginal and basal veins) can be used in the high-throughput protocol [81], as is the case with DNA barcoding. The high-throughput is an advantage over traditional identification methods, which are slow and require a high level of expertise [32]. The ability of DNA barcoding to discriminate between different species is reflected in the low intraspecific variation and relatively high interspecific variation [33]. In this study, the COI sequence variation analysis, together with the traditional methods of a PCA and biplot analysis, was useful in identifying the morphological cryptic complexes within *T. laeviceps s.l*. In addition, further analyses such as the multivariate ratio analysis (MRA) based on the insect body ratio [81,82] and geometric morphometrics (GM) based on the insect body outline [83] could be conducted in the future to confirm the results of the present study.

## 5. Conclusions

This study concludes that the combination of DNA barcoding for species identification and phylogenetic analysis, as well as traditional methods based on morphological grouping by body size and body coloration, can be reliably used to evaluate intraspecific variations such as the possible occurrence of subspecies within *T. laeviceps s.l*. The most significant morphological characteristics of *T. laeviceps s.l.* for distinguishing between groups or possibly subspecies were body size (FWL and FWW) and body coloration (TC and ASC). Furthermore, there were relatively high bootstrap values and genetic distances within the phylogenetic relationships of *T. laeviceps s.l*. These led to the separation of three distinct haplotype groups (12 out of 36 haplotypes) and suggest the existence of intraspecific cryptic speciation that warrants further investigation.

## Figures and Tables

**Figure 1 insects-14-00438-f001:**
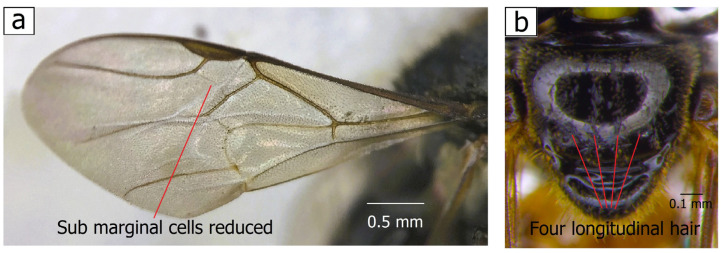
(**a**) Forewing: submarginal cells reduced; (**b**) thorax (dorsal): mesoscutum has four longitudinal hairs and three glabrous areas.

**Figure 2 insects-14-00438-f002:**
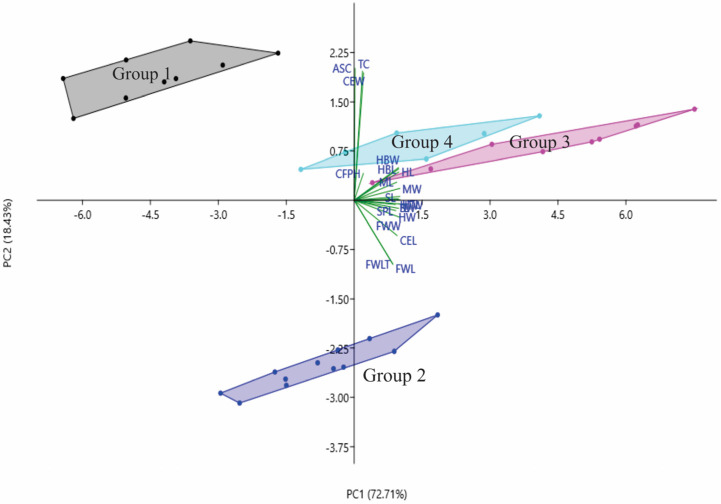
Principal component analysis biplot between *T. laeviceps s.l.* groups. Note: *T. laeviceps s.l.* Group 1 (TL 5 and TL 6); Group 2 (TL 1, TL 2, TL 3 and TL 4); Group 3 (TL 11 and TL 12) and Group 4 (TL7, TL8 and TL9). PCA group parameters: (A) (TL, HW, HL, CEL, FWL, ML, MW, SW, SL, HTL, HTW, HBL, HBW, CFPH and SPL); (B) (CEW, ASC and TC) and (C) (FWLT and FWW).

**Figure 3 insects-14-00438-f003:**
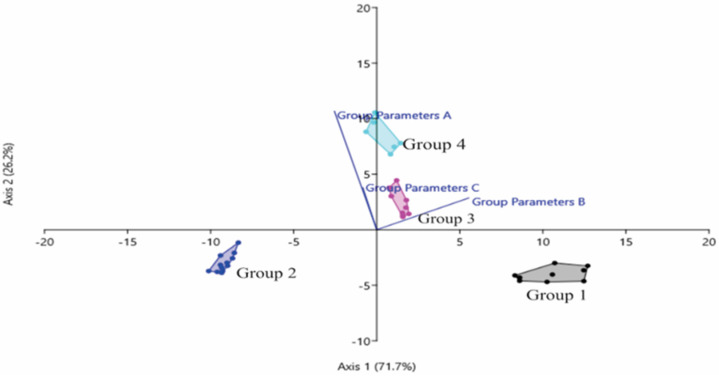
Linear Discriminant Analysis (LDA) biplot between groups of *T. laeviceps s.l*. Note: *T. laeviceps s.l.* Group 1 (TL 5 and TL 6); Group 2 (TL 1, TL 2, TL 3 and TL 4); Group 3 (TL 11 and TL 12) and Group 4 (TL7, TL8 and TL9). LDA group parameters: (A) (TL, HW, HL, CEL, FWL, ML, MW, SW, SL, HTL, HTW, HBL, HBW, CFPH and SPL); (B) (CEW, ASC and TC) and (C) (FWLT and FWW).

**Figure 4 insects-14-00438-f004:**
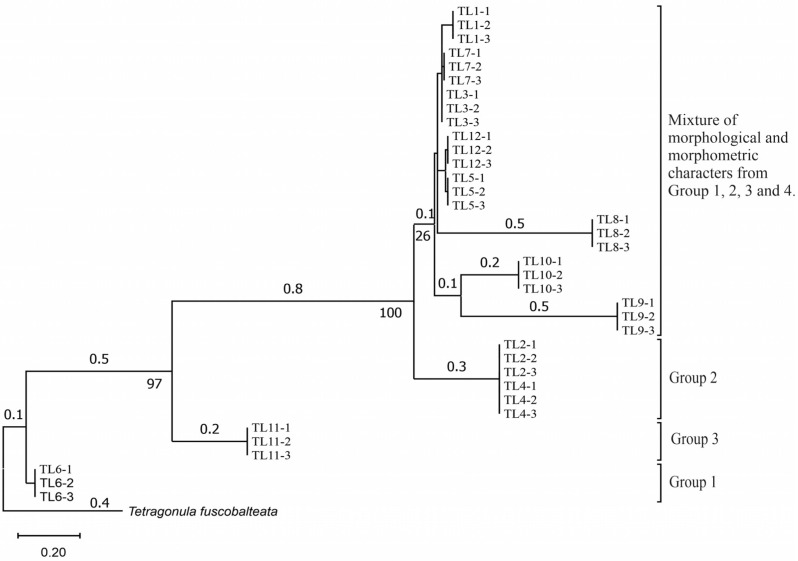
Phylogeny relationships of 36 *T. laeviceps s.l.* haplotypes and outgroup of mitochondrial COI genes. Node supports inferred from the bootstrap value for the maximum likelihood.

**Table 1 insects-14-00438-t001:** Four groups based on morphological distinctions, localities and gene codes of specimens of *T. laeviceps s.l.* collected in Sabah.

Group	Localities	Coordinates	Total Samples	References Collection No.	Sample Code and GenBank Accession No.	Hindleg
1	Kota Marudu	6.1763° N, 116.2328° E	6	TL5-1, TL5-2, TL5-3	OM935867	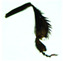
OQ545570	HTC = Black
OQ545571	BSC = Black
Tuaran	6.4958° N, 116.7610° E	TL6-1, TL6-2, TL6-3	OM935874	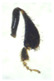
OQ545574	HTC = Black
OQ545575	BSC = Black
2	Kinarut	5.8370° N, 116.0433° E	12	TL1-1, TL1-2, TL1-3	OM944002	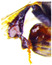
OQ538430	HTC = Dark Brown
OQ538429	BSC = Dark Brown
TL2-1, TL2-2, TL2-3	OM935876	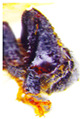
OQ545566	HTC = Dark Brown,
OQ545567	BSC = Dark Brown
Putatan	5.8912° N, 116.0485° E	TL3-1, TL3-2, TL3-3	OM935869	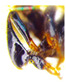
OQ545568	HTC = Dark Brown
OQ545569	BSC = Dark Brown
TL4-1, TL4-2, TL4-3	OM935866	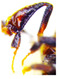
OQ545573	HTC = Dark Brown
OQ545572	BSC = Dark Brown
3	FSA	5.9306° N, 118.0116° E	9	TL10-1	OM935871	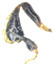
TL10-2	OQ545582	HTC = Blackish- Brown
TL10-3	OQ545587	BSC = Blackish- Brown
TL11-1	OM935873	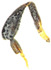
TL11-2	OQ545583	HTC = Blackish- Brown
TL11-3	OQ545585	BSC = Blackish- Brown
TL12-1	OM943485	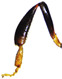
TL12-2	OQ545584	HTC = Blackish- Brown
TL12-3	OQ545586	BSC = Blackish- Brown
4	Tuaran	6.1763° N, 116.2328° E	9	TL7-1, TL7-2, TL7-3	OM935875	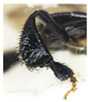
OQ545580	HTC = Black
OQ545577	BSC = Black
TL8-1, TL8-2, TL8-3	OM977030	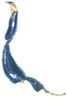
OQ545579	HTC = Black
OQ545578	BSC = Black
RDC	5.8760° N, 117.9449° E	TL9-1, TL9-2, TL9-3	OM977031	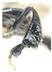
OQ545581	HTC = Black,
OQ545576	BSC = Black
Outgroup species	1	*Tetragonula fuscobalteata*	ON458746	-

Note: HTC = Hind tibia color, BSC = Hind basitarsus color.

**Table 2 insects-14-00438-t002:** Morphological characteristics and morphometric parameters of *T. laeviceps s.l*.

No.		Body Characteristics
1.	Body size	Total length (TL)
2.		Head width (HW)
3.		Head length (HL)
4.		Compound eye length (CEL)
5.		Compound eye width (CEW)
6.		Forewing length (including tegula) (FWLT)
7.		Forewing width (FWW)
8.		Forewing length (FWL)
9.		Mesoscutum length (ML)
10.		Mesoscutum width (MW)
11.		Mesoscutellum width (SW)
12.		Mesoscutellum length (SL)
13.		Hind tibia length (HTL)
14.		Hind tibia width (HTW)
15.		Hind basitarsus length (HBL)
16.		Hind basitarsus width (HBW)
17.	Body color and pubescence	Head color (HC)
18.		Clypeus color (CC)
19.		Antennae scape color (ASC)
20.		Clypeus and frons plumose pubescence (CFPP)
21.		Hind tibia color (HTC)
22.		Hind basitarsus color (BSC)
23.		Hind leg setae pubescence (LSP)
24.		Hind leg setae pubescence length (LSPL)
25.		Thorax mesoscutellum pubescence (SP)
26.		Thorax mesoscutellum pubescence length (SPL)
27.		Thorax color (TC)

**Table 3 insects-14-00438-t003:** Mean values for the body measurements (mm) of *T. laeviceps s.l.*

No.	Body Measurements	Group	N	Mean (mm)	Standard Error (SE)	Minimum (mm)	Maximum (mm)
1	TL ***	1	9	3.43 ^a^	0.01	3.39	3.47
		2	12	3.70 ^b^	0.01	3.65	3.75
		3	9	4.24 ^b^	0.12	3.73	4.52
		4	6	3.74 ^c^	0.02	3.69	3.79
2	HW ***	1	9	1.54 ^a^	0.01	1.5	1.59
		2	12	1.65 ^b^	0.01	1.6	1.7
		3	9	1.82 ^b^	0.04	1.64	1.92
		4	6	1.66 ^c^	0.02	1.6	1.71
3	HL ***	1	9	1.38 ^a^	0.01	1.33	1.43
		2	12	1.41 ^a^	0.01	1.36	1.45
		3	9	1.55 ^b^	0.02	1.43	1.61
		4	6	1.47 ^c^	0.01	1.44	1.5
4	CEL ***	1	9	1.03 ^a^	0.01	1	1.07
		2	12	1.20 ^b^	0.01	1.16	1.26
		3	9	1.28 ^c^	0.01	1.23	1.32
		4	6	1.25 ^c^	0.02	1.2	1.31
5	CEW ***	1	9	0.45 ^b^	0.01	0.41	0.49
		2	12	0.35 ^a^	0.01	0.3	0.41
		3	9	0.46 ^b^	0.01	0.4	0.51
		4	6	0.43 ^b^	0.01	0.4	0.46
6	FWLT ***	1	9	3.69 ^a^	0.01	3.65	3.75
		2	12	4.34 ^b^	0.01	4.29	4.38
		3	9	4.39 ^c^	0.01	4.32	4.45
		4	6	4.34 ^b^	0.02	4.28	4.39
7	FWW ***	1	9	1.21 ^a^	0.01	1.17	1.26
		2	12	1.37 ^b^	0.01	1.32	1.43
		3	9	1.52 ^c^	0.02	1.4	1.6
		4	6	1.42 ^b^	0.02	1.35	1.48
8	FWL ***	1	9	3.30 ^a^	0.01	3.26	3.35
		2	12	3.94 ^b^	0.01	3.89	3.98
		3	9	3.99 ^c^	0.01	3.92	4.05
		4	6	3.94 ^b^	0.02	3.89	3.99
9	ML ***	1	9	0.95 ^a^	0.01	0.91	0.99
		2	12	0.98 ^a^	0.01	0.94	1.04
		3	9	1.05 ^b^	0.01	1	1.13
		4	6	1.05 ^b^	0.03	0.97	1.14
10	MW ***	1	9	1.06 ^a^	0.01	1.02	1.11
		2	12	1.10 ^a^	0.01	1.05	1.14
		3	9	1.18 ^c^	0.02	1.1	1.25
		4	6	1.14 ^b^	0.01	1.1	1.18
11	SW ***	1	9	0.66 ^a^	0.01	0.62	0.71
		2	12	0.71 ^b^	0.01	0.66	0.75
		3	9	0.78 ^c^	0.01	0.72	0.83
		4	6	0.74 ^b^	0.02	0.68	0.79
12	SL ***	1	9	0.21 ^a^	0.01	0.18	0.27
		2	12	0.25 ^a^	0.01	0.2	0.29
		3	9	0.31 ^a^	0.02	0.23	0.38
		4	6	0.24 ^b^	0.02	0.19	0.29
13	HTL ***	1	9	1.36 ^a^	0.01	1.31	1.42
		2	12	1.50 ^b^	0.01	1.46	1.54
		3	9	1.69 ^c^	0.03	1.53	1.79
		4	6	1.57 ^d^	0.01	1.54	1.6
14	HTW ***	1	9	0.42 ^a^	0.01	0.38	0.48
		2	12	0.49 ^b^	0.01	0.44	0.53
		3	9	0.59 ^c^	0.01	0.52	0.65
		4	6	0.54 ^d^	0.02	0.5	0.6
15	HBL ***	1	9	0.55 ^a^	0.01	0.52	0.6
		2	12	0.57 ^a^	0.01	0.52	0.63
		3	9	0.66 ^c^	0.01	0.6	0.71
		4	6	0.62 ^b^	0.01	0.6	0.66
16	HBW ***	1	9	0.25 ^a^	0.01	0.21	0.31
		2	12	0.27 ^a^	0.01	0.23	0.33
		3	9	0.33 ^b^	0.01	0.3	0.38
		4	6	0.31 ^b^	0.02	0.25	0.36

Note: *** = significantly different at *p* < 0.001, ^a, b, c^ and ^d^ = show significant differences measured by Tukey b pairwise comparison. Groups: 1 (TL5-1, TL5-2, TL5-3, TL6-1, TL6-2 and TL6-3); 2 (TL1-1, TL1-2, TL1-3, TL2-1, TL2-2, TL2-3, TL3-1, TL3-2, TL3-3, TL4-1, TL4-2 and TL4-3); 3 (TL10-1, TL10-2, TL10-3, TL11-1, TL11-2, TL11-3, TL12-1, TL12-2 and TL12-3) and 4 (TL7-1, TL7-2, TL7-3, TL8-1, TL8-2, TL8-3, TL9-1, TL9-2 and TL9-3). Morphological characteristics: TL = total length, HW = head width, HL = head length, CEL = compound eye length, CEW = compound eye width, FWLT = forewing length (including tegula), FWW = forewing width, FWL = forewing length, ML = mesoscutum length, MW = mesoscutum width, SW = mesoscutellum width, SL = mesoscutellum length, HTL = hind tibia length, HTW = hind tibia width, HBL = hind basitarsus length and HBW = hind basitarsus width.

**Table 4 insects-14-00438-t004:** Mean rank and standard deviation (SD) of *T. laeviceps s.l.* body color and hairiness.

No	Body Color and Hair	Group	N	Mean Rank	Standard Error (SE)
1	HC ***	1	9	1.00	0.00
		2	12	2.00	0.00
		3	9	1.00	0.00
		4	6	1.00	0.00
2	CC ***	1	9	1.00	0.00
		2	12	2.00	0.00
		3	9	1.00	0.00
		4	6	1.00	0.00
3	ASC ***	1	9	2.33	0.17
		2	12	1.00	0.00
		3	9	2.00	0.00
		4	6	2.00	0.00
4	CFPP ***	1	9	1.00	0.00
		2	12	1.00	0.00
		3	9	1.00	0.00
		4	6	2.00	0.00
5	HTC ***	1	9	3.00	0.00
		2	12	2.00	0.00
		3	9	1.67	0.33
		4	6	1.00	0.00
6	BSC ***	1	9	3.00	0.00
		2	12	2.00	0.00
		3	9	1.67	0.33
		4	6	1.00	0.00
7	SP *	1	9	1.33	0.17
		2	12	1.00	0.00
		3	9	1.00	0.00
		4	6	1.00	0.00
8	LSPL ^NS^	1	9	1.33	0.17
		2	12	1.00	0.00
		3	9	1.33	0.17
		4	6	1.00	0.00
9	SP *	1	9	1.33	0.17
		2	12	1.00	0.00
		3	9	1.00	0.00
		4	6	1.00	0.00
10	SPL *	1	9	1.33	0.17
		2	12	1.00	0.00
		3	9	1.00	0.00
		4	6	1.00	0.00
11	TC ***	1	9	3.00	0.00
		2	12	2.00	0.00
		3	9	3.00	0.00
		4	6	3.00	0.00

Note: * = significantly different at *p* < 0.05, *** = significantly different at the *p* < 0.001 and NS = not significantly different. Groups: 1 (TL5-1, TL5-2, TL5-3, TL6-1, TL6-2 and TL6-3); 2 (TL1-1, TL1-2, TL1-3, TL2-1, TL2-2, TL2-3, TL3-1, TL3-2, TL3-3, TL4-1, TL4-2 and TL4-3); 3 (TL10-1, TL10-2, TL10-3, TL11-1, TL11-2, TL11-3, TL12-1, TL12-2 and TL12-3) and 4 (TL7-1, TL7-2, TL7-3, TL8-1, TL8-2, TL8-3, TL9-1, TL9-2 and TL9-3). Morphological characteristics: HC = head color, CC = clypeus color, ASC = antennae scape color, CFPP = clypeus and frons plumose pubescence, HTC = hind tibia color, BSC = basitarsus color, LSP = leg setae pubescence, LSPL = leg setae pubescence length, SP = mesoscutellum pubescence, SPL = mesoscutellum pubescence length and TC = thorax color.

**Table 5 insects-14-00438-t005:** Head and thorax morphology of *T. laeviceps s.l*.

Photographs of *T. laeviceps s.l.*		Group, Haplotypes	Morphological and Morphometrical Characteristics
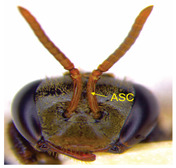	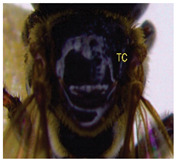	Group 2, TL2-1 and TL4-1	Antennae scape color (ASC) = yellowish-brown, Thorax color (TC) = dark brown. Body total length (TL) = 3.65 mm, Forewing length (FWL) = 3.89 mm. Forewing width (FWW) = 1.32 mm.
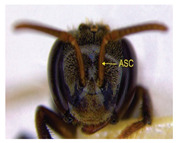	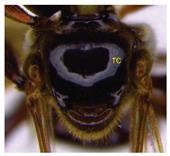		Antennae scape color (ASC) = yellowish-brown, Thorax color (TC)= dark brown. Body total length (TL) = 3.75 mm. Forewing length (FWL) = 3.98 mm. Forewing width (FWW) = 1.43 mm.
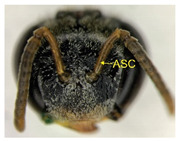	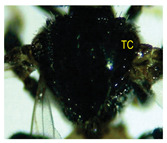	Group 3, TL11-1	Antennae scape color (ASC) = dark brown, Thorax color (TC) = black. Body total length (TL) = 4.52 mm. Forewing length (FWL) = 4.05 mm. Forewing width (FWW) = 1.60 mm.
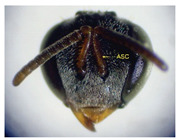	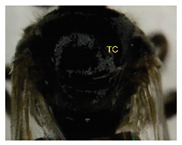	Group 1, TL 6-1	Antennae scape color (ASC) = blackish-brown, Thorax color (TC) = black. Body total length (TL) = 3.39 mm. Forewing length (FWL) = 3.26 mm. Forewing width (FWW) = 1.17 mm.

## Data Availability

Data is contained within the article.

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
