# Peer review of "Phylogenetic and Morphological Characteristics Reveal Cryptic Speciation in Stingless Bee, Tetragonula laeviceps s.l. Smith 1857 (Hymenoptera; Meliponinae)"

_insects, 2023, doi:10.3390/insects14050438_

Round 1

Reviewer 1 Report

Dear authors,

This is an interesting work that contributes to the understanding of cryptic stingless bee species. However, it needs more attention before being accepted. I list below some suggestions and comments to improve the quality of the paper:

1 - Page 1, lines 13 and 14: This sentence is not elaborated in English. Please rewrite it.

2 - Page 1, line 17: you mention three groups within T. laeviceps. Who are these three groups? Three morphological patterns already known or that you identified in this study?

3 - Page 1, line 30: you mention the three groups again and yet I still have doubts about who these three groups are. Please clarify.

4 - Page 2, line 47: change "...in having.." to "...by having...".

5 - Page 2, line 48: "longer wing length". Longer than? this sentence is incomplete.

6 - Page 2, line 48: "hind end". Would that be the metasoma? If yes, please change the spelling.

7 - Page 2, line 49: "setae [1]. There are". Chanfe for "setae and by having the hind wing with five hamuli".

8 - Page 2, line 52: T. iridipennis is cited for the first time, please review the rules of this Journal to verify if it is necessary to include the author and year.

9 - Page 2, line 53: "renamed T...". Please add "as" between the words.

10 - Page 2, line 62: hindwing --> hind wing.

11 - Page 2, lines 72 and 73: head and antennae and legs --> head, antennae and legs.

12 - Page 3, line 97: what is the collection method used? Manual collection? Malaise? Attractive trap?

13 - Page 3, Table 1: Only now I understand what the three groups are, so you need to improve the information about them in the previous parts of the MS.

14 - Page 3, line 108: Please remove the underscore from the species name.

15 - Page 3, line 110: follows; --> follows:

16 - Page 3, line 117: were measured; --> were measured:

17 - Page 3, lines 117 to 121: measures are generally complex and need to be well explained in the text to enable the study to be replicated. For example: was head length measured from the median region? from the furthest points of the head? I strongly suggest that a plate with images demonstrating how each measurement was performed, or that right after each measurement, it is explained in detail how it was measured.

18 - Page 3, lines 126: about hairiness, what are the observed states? Presence and absence? Very hairy and scarce? if yes, you need to explain how each state was ranked.

19 - Page 4, line 134: and 1 --> and one.

20 - Page 4, lines 142 and 143: "For more details on the DNA extraction procedure, amplification of the PCR products and PCR purification". I think this sequence is incomplete, no?

21 - Page 5 (Sequence Editing and Alignment): Why didn't BLAST the Genbank to confirm that the sequences are correct?

22 - Page 5, lines 152 and 153: Please insert a space between the title and the text above.

23 - Page 5, line 154: Please italize the species name.

24 - Page 5, line 171: Why only one terminal as an outgroup? We typically use outgroups to improve analysis.

25 - Page 5, lines 170 and 171: How did you choose K2P?

26 - Page 5, lines 171 and 172: what model was obtained?

27 - Page 5, line 175: Please italize the species name.

28 - Page 5, line 184: 3 glabrous --> three glabrous.

29 - Page 6, Figure 1: The images need to be improved, for example, it's hard to see the four lines of hair.

30 - Page 6, line 194: Please insert the letter A and the letter B, one for each image.

31 - Page 6, line 194: 4 longitudinal --> four longitudinal.

32 - Page 6, line 195: 3 glabrous --> three glabrous.

33 - Page 7, lines 198, 200, 221 and 225: Please italize the species name.

34 - Page 8, lines 236, 241, 247 and 257: Please italize the species name.

35 - Page 8, line 248: "with a size of 623 bp". Was this size obtained for all 12 samples or is it an average?

36 - Page 9, lines 262, 263, 265 and 276: Please italize the species name.

37 - Page 9, lines 265 and 266: "Figure 2, Clade I appear" --> Figure 2, the Clade I appear".

38 - Page 9, lines 273 and 274: Normally, clades are groupings that include a common ancestor and all its descendants. The "clade IV" has only one terminal, so I consider it more appropriate to say that it has only one lineage, since there is no grouping to be considered a clade.

39 - Page 10, lines 289, 290, 313 and 329: Please italize the species name.

40 - Page 11, lines 345, 361, 364, 365, 371, 379 and 380: Please italize the species name.

41 - Page 12, lines 387, 390, 395, 398, 403, 409, 412, 421, 429 and 434: Please italize the species names and check throughout the text that all species names are highlighted as they should.

42 - Page 13, lines 449, 450 and 457: Please italize the species name.

43 - Page 13, Acknowledgments: Please include acknowledgments to the editor and reviewers.

44 - Pages 13 and 14, References: I have not reviewed all the references, but it is noticeable that they are out of standard, for example, in some the Journal name is in all capital letters, in others abbreviated. Please review carefully following the instructions to the authors.

All the best,

Author Response

Please see the attachment for the response to reviewer 1 comments.

Author Response

Please see the attachment for the response to reviewer 2 comments.

Reviewer 3 Report

Dear Prof. Benedick,

I have carefully read your manuscript entitled "Phylogenetic and Morphological Characteristics Reveals Cryptic Speciation in Stingless Bee, Tetragonula laeviceps (Hymenoptera; Meliponinae)". The paper contains new information on morphological and molecular traits of T. laeviceps complex and therefore it could be considered for publication in Insects. Nevertheless, I have to recommend rejection of the manuscript due to several reasons. First, it seems that the authors do not clearly understand the main goal of the study, i.e., they treat four clades within this complex as potential "subspecies". However, high genetic distances between these clades strongly suggest the existence of several cryptic species. Moreover, some bootstrap values in the existing molecular analysis of this complex are too low, i.e., below 50%, to reliably support particular clades. In addition, a diagram showing the results of PCA analysis, could clearly demonstrate the existence of morphological groupings within the studied complex. Furthermore, I would also recommend using MRA to highlight most important morphometric differences between these groupings (see https://doi.org/10.1093/sysbio/syr061 and https://doi.org/10.1111/syen.12081). Among other organisms, this approach was successfully used to analyze morphological differences between closely related bee species (https://doi.org/10.1007/s13127-021-00513-z). 

Author Response

Please see the attachment for the response to reviewer 3 comments.

Round 2

Reviewer 3 Report

Dear Prof. Benedick,

I have carefully read a revised version of your manuscript. As you must remember, I have rejected the previous version of the paper due to numerous theoretical and methodological problems. However, as far as I can see, most of these problems still persist, and some new ones have arisen. For example, certain photos suggest that various individuals strongly differ in the body proportions, and therefore can hardly belong to the same morphospecies. Moreover, you do not cite any of the papers which I previously suggested, etc. etc. I thus have to finally reject your manuscript.
